# Impact of Small Intestinal Bacterial Overgrowth in Patients with Inflammatory Bowel Disease and Other Gastrointestinal Disorders—A Retrospective Analysis in a Tertiary Single Center and Review of the Literature

**DOI:** 10.3390/jcm12030935

**Published:** 2023-01-25

**Authors:** Julia Wanzl, Katharina Gröhl, Agnieszka Kafel, Sandra Nagl, Anna Muzalyova, Stefan Karl Gölder, Alanna Ebigbo, Helmut Messmann, Elisabeth Schnoy

**Affiliations:** Internal Medicine III, University Hospital Augsburg, 86156 Augsburg, Germany

**Keywords:** Crohn’s disease, ulcerative colitis, small intestinal bacterial overgrowth, inflammatory bowel disease

## Abstract

Background: Small intestinal bacterial overgrowth (SIBO) is often found in patients with gut dysbiosis such as irritable bowel syndrome. Recently, the association of SIBO and inflammatory bowel disease (IBD) has been described in some cases. While clinical symptoms might be similar in IBD and SIBO, treatment is quite different for both diseases. Therefore, the differentiation between SIBO or a flare in IBD patients is key to optimizing treatment for these patients. Methods: We retrospectively investigated our patients with IBD receiving a glucose breath test for SIBO and correlated the results with the clinical symptoms (clinical remission or active disease). Results: 128 patients with the diagnosis “colitis” were analyzed in our cohort. Fifty-three (41.4%) patients had Crohn’s disease and 22 (17.2%) patients were suffering from ulcerative colitis. Seventy-four (57.8%) were female and 54 (42.2%) were male patients. A total of 18 (14.1%) patients had a positive testing for SIBO. Eleven (61.1%) cases were associated with CD patients and two (11.1%) with UC. IBD patients in clinical remission had a positive SIBO in six (19.4%) cases, while IBD patients with active disease were positive in nine (15.3%) cases. The proportion of positive SIBO in active IBD patients was higher; however, it did not reach significance. Older age was a risk factor for SIBO in patients with CD (*p* < 0.003). Conclusions: In our study, we could show that an increased amount of SIBO was found in IBD patients and was especially more frequent in patients with CD than in those with UC. In UC patients, SIBO rates were not different to patients with other gastrointestinal diseases investigated (e.g., infectious colitis, collagenous colitis, or irritable bowel syndrome). In active IBD, positive SIBO was detected more often numerically compared to quiescent disease; however, due to the low number of patients included, it was not significant. However, older age was a significant risk factor for SIBO in patients with CD. SIBO is of clinical relevance in the vulnerable patient cohort with IBD, and its real prevalence and impact needs to be investigated in further and larger clinical trials.

## 1. Introduction

Small intestinal bacterial overgrowth (SIBO) is characterized by the presence of excessive bacteria in the small intestine. SIBO can be associated with a variety of different clinical conditions and is often found in patients with irritable bowel syndrome (IBS) [1]. Recently, the presence of SIBO was described in patients with inflammatory bowel disease (IBD) [2,3,4].

Clinical symptoms can be nonspecific, such as abdominal pain, changes in stool frequency and formation, diarrhea or bloating, and psychological disorders therefore also mimicking a flare of IBD. Simultaneously, there is no diagnostic gold standard for SIBO. As a consequence, the true prevalence and connection to other diseases still remain unclear and is definitely underrecognized so far.

In diseases with gut stasis such as constipation predominant IBS, the prevalence of SIBO is high [1,3].

IBD includes both ulcerative colitis (UC) and Crohn’s disease (CD). Both diseases occur with periods of activity interrupted by times of complete remission [5]. Despite remission and disease control, many patients with IBD still suffer from symptoms mimicking irritable bowel syndrome with consecutive reduced quality of life [6,7,8,9]. According to the literature, about 40% of patients with IBD suffer from IBS-associated symptoms [9,10].

Not only in gastrointestinal disorders such as IBD but also in patients with liver diseases, an increased prevalence of SIBO is found compared to control patients. In these patients, SIBO is found independently of the stage of the liver disease (e.g., early or advanced liver disease) [11]. The use of proton pump inhibitors was found to be another risk factor for SIBO [12].

Different diets and supplements have been shown to improve the symptoms of IBS patients as well as IBD patients with IBS symptoms (e.g., low-FODMAP, vitamin D, or probiotics) [13,14,15,16,17,18]. Treatment of SIBO includes antibiotics, probiotics, and prokinetics, while in recent years therapy with rifaximin is regarded as the standard of care for SIBO [12]. The impact of SBIO in IBD patients still remains unclear, while clinical symptoms are similar for both diseases. Therefore, it is important to differentiate between patients with IBD having a flare from those having SIBO because treatments are completely different for both diseases.

In our study, we investigated the prevalence and impact of SIBO in our patient cohort, with a focus on patients with both UC and CD in accordance with their IBD activity status (remission or active disease) and their clinical symptoms and provide a review of the literature for this topic.

## 2. Materials and Methods

### 2.1. Study Design

A single center retrospective analysis was performed at the University hospital Augsburg, Germany. Data were obtained from all patients presenting at our local department for functional diagnostics between 1 January 2010 and 31 December 2021. All adult patients receiving any breath test were included in the initial cohort. Thereafter, patients with the diagnosis of “colitis” (CD, UC or indeterminate colitis, infectious colitis, and others (e.g., IBS, collagenous colitis, or unknown)) and having a glucose breath test (Stimotron, Wendelstein, Germany) for SIBO as recommended [19] were filtered and included in further analysis. Patients with IBD under clinical remission (absence of clinical activity signs) or with active disease were permitted. Patients receiving antibiotic treatment or proton pump inhibitors were not included in our study.

Clinical data were obtained from the patient’s report at the time of the breath test and when needed at the charts, or the patient’s documentation at the time of testing. Data collection and retrospective analysis of patient information were anonymized in accordance with the Declaration of Helsinki.

In our center, patients need to fast for 12 h before testing; smoking is also not permitted within this time frame. During the breath test, patients received a predefined amount of a carbohydrate substrate to drink (75 g glucose in 400 mL drinking water). This is thereafter metabolized when exposed to the gastrointestinal microbes, producing hydrogen and methane. These substrates can be absorbed and exhaled through the lungs, collected, and analyzed. In our hospital, glucose is the regular substrate to be tested. Tests were regarded to be positive when initial levels were augmented by 20 ppm. The study was conducted in accordance with the Declaration of Helsinki, and the protocol was approved by the Ethics Committee of Regensburg.

### 2.2. Statistical Evaluation

Data were initially analyzed descriptively with categorical variables presented as absolute frequencies and percentages. Continuous variables were described with means and standard deviations. Dependency between categorical variables was carried out using Fisher’s Exact Test. Multivariate analysis of the group was performed using binomial logistic regression. The IBM SPSS (Statistical Package for the Social Sciences) version 27 program was used for data analysis.

## 3. Results

### 3.1. Patients Demographics

One hundred twenty-eight patients with the diagnosis of “colitis” were included in the final analysis (54 (42.2%) men, 74 (57.8%) women). Mean age in the study population was 58.87 (SD = 18.696) years. Fifty-three (41.4%) patients were suffering from CD, 22 (17.3%) patients from UC, and 8 (6.3%) patients from indeterminate colitis. Six (4.7%) patients had infectious colitis, 39 (30.5%) patients had other colitis, 26 (20.3%) patients had irritable bowel syndrome, 3 (2.3%) patients had collagenous colitis, and 10 (7.8%) were of unknown origin). Patient demographics are given in Table 1.

### 3.2. Positive Testing for SIBO

In 18 of 128 cases (14.1%), positive testing for SIBO was found. Eleven (61.1%) cases of CD were detected, along with two (11.1%) of UC, one (5.6%) of indeterminate colitis, one (5.6%) of infectious colitis, two (11.1%) of irritable bowel syndrome, and one (5.6%) of short bowel syndrome. Results are given in Table 2. Current medication for patients with IBD and positive SIBO are also given in Table 2.

### 3.3. IBD Activity

Of all IBD patients, 31 (24.2%) patients were in clinical remission; however, of those, six (19.4%) patients were positively tested for SIBO. A slightly smaller share of 59 patients not in clinical remission tested positive (*p* = 0.767, 9 (15.3%) for SIBO. Clinical complaints were quite heterogeneous (weight loss, *n* = 3; diarrhea, *n* = 6; abdominal pain or bloating, *n* = 3, multiple choices possible) in both groups (Table 3). Patients with a positive SIBO test in accordance with the underlying disease (CD and UC) were more frequent, but not significantly (*p* = 0.636) (Table 4).

### 3.4. Effect of Age on Positive SIBO

When comparing patients with a positive SIBO test in accordance with the underlying disease (patients with CU, UC, and other diagnosis), results between groups did not reach statistical significance (*p* = 0.192); however, in a multivariate analysis of the risk factors, patients with CD and older age had a significant risk of SIBO (Table 5 and Table 6).

## 4. Discussion

In our single center retrospective study, SIBO was more often found in patients with CD than in those with UC. Furthermore, SIBO was more often positive in patients with IBD when they were not in clinical remission. SIBO rates were similar in patients with UC, infectious colitis, and collagenous colitis, as well as those with IBS. Older age in patients with CD was associated with a significantly higher risk of SIBO.

The numerical higher rate of positive SIBO tests in 17.3% CD patients might be due to several predisposing factors in these patients. It is well known that, in general, patients with IBD suffer from gut dysbiosis [16] with a small intestinal bacterial overgrowth, mainly due to chronic intestinal inflammation in the small intestine, while in UC inflammation is mostly limited to the colon. In patients with IBS, small intestinal inflammation is a risk factor for SIBO [17].

In addition, ileocecal resection can lead to increased intestinal permeability due to the lack of the ileocecal valve and dysmotility [2,18,19,20]. Consequently, these factors might contribute to reduced digestion and absorption of nutrients and the production of osmotically active metabolites in the lumen, leading to clinical symptoms such as discomfort, bloating, and diarrhea [21,22]. In our cohort, patients were suffering from similar symptoms to those described in the literature, with mainly abdominal pain, diarrhea, and weight loss or bloating.

The exact prevalence of SIBO is unknown. Our incidences of positive SIBO, especially in CD, is in accordance with other studies published [23,24,25], while some studies report even higher rates of positive SIBO. Only in one meta-analysis was the prevalence of methane-positive SIBO in IBD patients lower, with 5.3% for CD as compared to those with UC (20.2%). In general, literature data indicate a frequency of overlap of SIBO in patients with inflammatory bowel disease of 18–30% for CD and 14–17.8% for UC [26,27]. In our cohort, we had similar results to those described in the literature for patients with CD, having a positive SIBO test in 21.1% of our cases. In contrast, positive SIBO was rather low in UC in our cohort (9.1%) compared to the literature; however, the rate of positive SIBO was similar for patients with IBS (7.7%).

IBD compromising CD and UC are clinical heterogeneous diseases. While patients with an acute flare suffer from symptoms such as abdominal pain, diarrhea, and weight loss, it is well known that patients with IBD in remission can still suffer from ongoing “flare like” symptoms despite adequate control of the inflammation in the gut [28,29]. SIBO can present with similar symptoms to a flare in IBD patients. However, treatment is different for both diseases [30]. Therefore, careful medical history and testing is key to differentiate an acute flare in IBD from other reasons such as SIBO. In general, non-invasive glucose or lactulose breath testing are methods to diagnose SIBO, while invasive testing is not practical in everyday routines [27]. A breath test is easy to offer to the patient, with a high acceptance, and should always be performed in IBD patients with suspicion of SIBO.

In the treatment of SIBO, rifaximin is effective and safe and is associated with a rapid improvement of gastrointestinal symptoms [31]. So far, evidence of specific diets in the treatment of SIBO is still rare, resulting in no clear dietician recommendations [32], e.g., low-FODMAP diets might help to improve symptoms of SIBO. However, recurrence rates of SIBO after treatment are still high [33]. Concerning further treatment, the standard of care in our hospital is treatment with rifaximin as recommended [34,35,36,37]. However, due to its retrospective character, further treatment could not be analyzed for all patients included.

In our study, we could show that older patients with CD have a higher risk for SIBO. So far, to our knowledge, no other study could show older age as a risk factor for SIBO in IBD. In one meta-analysis, older age was also a risk factor for SIBO in patients with IBS [38], which was confirmed in another study [39].

Our study has some limitations. First, its retrospective analysis, regarding the current clinical symptoms of all IBD patients provided at the time of testing, has no objective confirmation of the current activity status in IBD patients (e.g., missing biomarker or endoscopy). Furthermore, due to the limited number of patients, the statistical analysis was not elusive for all questions raised.

In addition, concurrent medication or former surgery as potential risk factors in patients with IBD were not included in our analysis. In addition, patients were only screened for gastrointestinal disorders with a focus on IBD. Other concomitant diseases that might have additional influence on the diagnosis of SIBO, such as liver diseases, were not analyzed in our study [11].

## 5. Conclusions

In our study, we could clearly show that SIBO plays a role in IBD patients and other gastrointestinal disorders. Its prevalence in patients with IBD in remission might still be underestimated. Especially in patients with CD, SIBO is increasingly found in older age. Clinically, SIBO should be routinely considered in order to optimize the treatment of IBD patients.

## Figures and Tables

**Table 1 jcm-12-00935-t001:** Patient demographics of all patients with colitis: CD, Crohn’s disease; UC, ulcerative colitis; IBS, irritable bowel syndrome.

Dimension	Category	*n* (%)
Sex	male	54 (42.2%)
	female	74 (57.8%)
Diagnosis	CD	53 (41.4%)
	UC	22 (17.3%)
	indeterminate colitis	8 (6.3%)
	infectious colitis	6 (4.7%)
	IBS	26 (20.3%)
	collagenous colitis	3 (2.3%)
	unknown	10 (7.8%)

**Table 2 jcm-12-00935-t002:** Patients with positive testing for SIBO are provided in detail in this table.

**Dimension**		**Category**			***n* (%)**	
Positive SIBO,		CD			11 (61.1%)	
		UC			2 (11.1%)	
		indeterminate colitis			1 (5.6%)	
		infectious colitis			1 (5.6%)	
		IBS			2 (11.1%)	
		short bowel syndrome			1 (5.6%)	
		Total			18	
**Patients with IBD and positive testing for SIBO, and current medication**						
	patient	diagnosis	year of birth	SIBO	date of SIBO	treatment at time of positive SIBO
1	male	CD	1973	positive	2010	azathioprine budesonide
2	female	CD	1954	positive	2010	azathioprine prednisolone
3	female	CD	1968	positive	2012	no systemic therapy
4	male	CD	1955	positive	2013	azathioprine prednisolone
5	female	CD	1957	positive	2013	mesalazine budesonide
6	female	CD	1942	positive	2016	prednisolone
7	male	CD	1960	positive	2016	prednisolone azathioprine adalimumab
8	female	CD	1970	positive	2016	budesonide mesalazine azathioprine
9	female	CD	1963	positive	2018	ustekinumab
10	male	CD	1981	positive	2019	infliximab
11	female	CD	1952	positive	2020	ustekinumab
12	female	UC	1956	positive	2016	vedolizumab
13	male	UC	1966	positive	2019	mesalazine

**Table 3 jcm-12-00935-t003:** IBD patients with positive SIBO in accordance with their clinical activity status.

Dimension	Total	Positive SIBO		Negative SIBO	
	*n*	*n*	%	*n*	%
clinical remission	31	6	19.4%	25	83.9%
clinical activity	59	9	15.3%	50	86.4%

**Table 4 jcm-12-00935-t004:** Patients with positive SIBO test in accordance with the underlying disease, sex, and age.

Dimension	Total	SIBO Positive		SIBO Negative	
Disease Type	*n*	*n*	%	*n*	%
CD	52	11	21.2%	41	78.8%
UC	22	2	9.1%	20	90.9%
indeterminate colitis	8	1	12.5%	7	87.5%
infectious colitis	6	1	16.7%	5	83.3%
irritable bowel syndrome	26	2	7.7%	24	92.3%
collagenous colitis	3	0	0.0%	3	100.0%
other	10	1	10.0%	9	90.0%
age		18	66.389 (SD = 15.217)	109	57.93 (SD = 19.024)
female	74	10	13.5%	64	86.5%
Male	54	8	14.8%	46	85.2%

**Table 5 jcm-12-00935-t005:** Patients with positive SIBO test in accordance with the underlying disease comparing patients with CU, UC, and other diagnosis and age.

		CD		UC		Other			
		*n*	%	*n*	%	*n*	%	*n*	%
subgroup	SIBO positive	11	21.2%	2	9.1%	5	9.4%	18	14.2%
subgroup	SIBO negative	41	78.8%	20	90.9%	48	90.6%	109	87.4%
Total		52	100.0%	22	100.0%	53	100.0%	127	100.0%

**Table 6 jcm-12-00935-t006:** Summary of the risk factors associated with positive SIBO.

SIBO Positive	Significance	Exp(B)	95% Confidence Interval	
			Lower Limit	Upper Limit
age	0.004	1.077	1.024	1.133
CD	0.003	17.888	2.743	116.669
UC and IC	0.134	4.078	0.65	25.583

## Data Availability

The data presented in this study are available on request from the corresponding author.

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
