# Peer review of "Impact of Small Intestinal Bacterial Overgrowth in Patients with Inflammatory Bowel Disease and Other Gastrointestinal Disorders—A Retrospective Analysis in a Tertiary Single Center and Review of the Literature"

_jcm, 2023, doi:10.3390/jcm12030935_

Round 1
Reviewer 1 Report
Well written study on an interesting subject. However, I do believe that the title is misleading since the authors did not include only patients with IBD but patients with all sorts of bowel disorders. In fact approximately only 60% of patients included had IBD. Therefore, the authors should either analyze only those patients or modify the title of the manuscript.
Furthermore, I do believe that the authors should include information regarding the medications the IBD patients included were taking. This information is crucial since several medication could possibly alter the small intestinal flora of the patient. Several CD patients are for example receiving ciprofloxacin or metronidazole for their disease. Where these patients excluded? Although the number of patients is small and a statistical analysis of patients receiving antibiotics, corticosteroids or biologics is not possible, this should also be included in the limitations of the study.
Finally the authors should address their point of view regarding the management of patients with IBD found positive for SIBO. Did they administer rifaximin? Did they do something else?
Author Response
Response to reviewer 1
We really thank this reviewer very much for considering our manuscript entitled „ Impact of small intestinal bacterial overgrowth in patients with inflammatory bowel disease – a retrospective analysis in a tertiary single center and review of the literature “ for publication in Journal of Clinical Medicine after revision.
Please find attached
- a point-to-point reply to this reviewers' comments that addresses each comment of reviewer 1.
- a revised version of the manuscript with changes made in red (from reviewer 1 and 2)
- Well written study on an interesting subject.
- We thank this reviewer for this positive comment to our study.
- However, I do believe that the title is misleading since the authors did not include only patients with IBD but patients with all sorts of bowel disorders. In fact approximately only 60% of patients included had IBD. Therefore, the authors should either analyze only those patients or modify the title of the manuscript.
- We appreciate this comment and agree that our study focuses on patients with IBD but also other gastrointestinal disorders were included in our study. Therefore, as suggested, we modified the title of our manuscript. The new title includes the hint that not only IBD patients were included in our analysis and we hope: “Impact of small intestinal bacterial overgrowth in patients with inflammatory bowel disease and other gastrointestinal disorders – a retrospective analysis in a tertiary single center and review of the literature” (page 1).
- Furthermore, I do believe that the authors should include information regarding the medications the IBD patients included were taking. This information is crucial since several medication could possibly alter the small intestinal flora of the patient. Several CD patients are for example receiving ciprofloxacin or metronidazole for their disease. Where these patients excluded?
- We thank the reviewer for this point. In the revised version, a new table (marked in red, page 3-4, table 2B) was included. All patients with IBD and positive SIBO testing are provided with their current medication at the time of their breath test in this table. Antibiotic treatment was an exclusion criteria (beside patients with current antibiotic treatment do normally not get a glucose breath test in our clinic). This information was now added in the Material and Methods section (page 2).
- Although the number of patients is small and a statistical analysis of patients receiving antibiotics, corticosteroids or biologics is not possible, this should also be included in the limitations of the study.
- We thank the reviewer for this suggestion. We have now added this information in the Discussion and highlighted this point raised by the reviewer (page 7)
- Finally the authors should address their point of view regarding the management of patients with IBD found positive for SIBO. Did they administer rifaximin? Did they do something else?
- We thank the reviewer for this helpful comment. In our first version of the manuscript, treatment options for SIBO were already included. We have now added the information about how patients in our center are normally treated (standard of care is rifaximin). We also point out in our revised manuscript that no follow up of our patient cohort was performed. Therefore, we unfortunately cannot provide the treatment applied for each patient in our cohort (page 7).

Reviewer 2 Report
The topic is very interesting, but in its current form it does not bring any radical novelties. it seems that a few major changes could make the article more attractive and allow it to be published in JCM
I suggest expanding the short introduction to the topic
1. - SIBO has also been frequently documented in association with liver disease
Shah A, Shanahan E, Macdonald GA, Fletcher L, Ghasemi P, Morrison M, et al. Systematic review and meta-analysis: prevalence of small intestinal bacterial overgrowth in chronic liver disease. Semin Liver Dis. 2017;37:388–400 Reaffirms, within the limits of diagnostic methodology, the association between SIBO and chronic liver disease.
2. - describe recommendations for eating or avoiding foods in the introduction,
Kikut J, Konecka N, Ziętek M, Kulpa D, Szczuko M. Diet supporting therapy for inflammatory bowel diseases. Eur J Nutr. 2021 Aug;60(5):2275-2291. doi: 10.1007/s00394-021-02489-0
3. -Some studies have suggested an association between SIBO and the use of proton-pump inhibitors (PPIs)
Franco DL, Disbrow MB, Kahn A, et al. Duodenal aspirates for small intestine bacterial overgrowth: Yield, PPIs, and outcomes after treatment at a tertiary academic medical center. Gastroenterol Res Pract 2015;2015:971582.
4. - add a small paragraph on treatments - (e.g. ciprofloxacin, norfloxacin and metronidazole)
in materials and methods
5. - please specify the type of breath tests performed (glucose, lactulose), tests assert that the false positive rate for hydrogen breath tests is only 15% and lower for glucose than lactulose tests
Rezaie A, Buresi M, Rao S. Response to Paterson et al. Am J Gastroenterol. 2017;112:1889–92.
6. - principle of operation (company, city, state)
The North American consensus concluded that “a rise in hydrogen of ≥ 20 ppm
Rezaie A, Buresi M, Lembo A, Lin H, McCallum R, Rao S, et al. Hydrogen and methane-based breath testing in gastrointestinal disorders: the North American consensus. Am J Gastroenterol. 2017;112:775-8
7-- what were the exclusion criteria, maybe: proton pump inhibitors, opioids, gastric bypass, colectomy and motility disorders
8-Patients should also fast 8–12 hours before the breath test, avoid smoking the day of the breath test, and minimize physical exertion during the breath test
Results
9.-are there any patient data on the co-occurrence of diseases: liver, obesity, diabetes, ? what is their relationship to SIBO?
should be presented and discussed
discussion
10. Chen et al linked IBS with age in their meta-analysis, therefore increasing the impact of SIBO comorbidities is desirable
Chen B, Kim JJ, Zhang Y, Du L, Dai N. Prevalence and predictors of small intestinal bacterial overgrowth in irritable bowel syndrome: a systematic review and meta-analysis. J Gastroenterol. 2018. https://doi.org/10.1007/s00535-018-1476-9.
11. Choung et al found that SIBO appears to be more prevalent in women and in older individuals
Choung RS, Ruff KC, Malhotra A, et al. Clinical predictors of small intestinal bacterial overgrowth by duodenal aspirate culture. Aliment Pharmacol Ther 2011;33:1059–67.
12. Limitations of the study should be presented
13. there is no conclusion - a separate summary section should be added
Author Response
Response to reviewer 2
We really thank this reviewer very much for considering our manuscript entitled „ Impact of small intestinal bacterial overgrowth in patients with inflammatory bowel disease – a retrospective analysis in a tertiary single center and review of the literature “ for publication in Journal of Clinical Medicine after revision.
Please find attached
- a point-to-point reply to this reviewers' comments that addresses each comment of reviewer 2.
- a revised version of the manuscript with changes made in red (from reviewer 1 and 2)
- The topic is very interesting, but in its current form it does not bring any radical novelties. it seems that a few major changes could make the article more attractive and allow it to be published in JCM
- I suggest expanding the short introduction to the topic
- We thank this reviewer for this comment and for the possibility to upload a revised version of our manuscript. We have addressed all points by this reviewer carefully and we have therefore made major improvements to our study.
- - SIBO has also been frequently documented in association with liver disease
Shah A, Shanahan E, Macdonald GA, Fletcher L, Ghasemi P, Morrison M, et al. Systematic review and meta-analysis: prevalence of small intestinal bacterial overgrowth in chronic liver disease. Semin Liver Dis. 2017;37:388–400 Reaffirms, within the limits of diagnostic methodology, the association between SIBO and chronic liver disease.- We thank the reviewer for this point. We have added this important information to our manuscript, the details provided can be found on page 2 (marked in red) in the introduction. The provided literature was added to the manuscript.
- - describe recommendations for eating or avoiding foods in the introduction,
Kikut J, Konecka N, Ziętek M, Kulpa D, Szczuko M. Diet supporting therapy for inflammatory bowel diseases. Eur J Nutr. 2021 Aug;60(5):2275-2291. doi: 10.1007/s00394-021-02489-0- We thank this reviewer for this suggestion. Details on diets in different diseases included were already provided in our first version of the manuscript. We have not added more and detailed information in the introduction (page 2) on this topic. This topic is again addressed in the discussion (page 7). The recommended literature was added to the new version of the manuscript.
- -Some studies have suggested an association between SIBO and the use of proton-pump inhibitors (PPIs)
Franco DL, Disbrow MB, Kahn A, et al. Duodenal aspirates for small intestine bacterial overgrowth: Yield, PPIs, and outcomes after treatment at a tertiary academic medical center. Gastroenterol Res Pract 2015;2015:971582.- We appreciate this feedback. PPIs are one of the risk factors for SIBO. We have now added this information to the introduction (page 2, red). In addition, this was of course one of our exclusion criteria (as well as antibiotic treatment). We have now improved our Material an Methods section and have included this important point (page 2).
- - add a small paragraph on treatments - (e.g. ciprofloxacin, norfloxacin and metronidazole)
- We thank the reviewer for this point. In our revised manuscript, we provide more information on this point in both introduction and discussion. We hope that we have now sufficiently addressed this important topic (page 2 and 7).
in materials and methods
- - please specify the type of breath tests performed (glucose, lactulose), tests assert that the false positive rate for hydrogen breath tests is only 15% and lower for glucose than lactulose tests
Rezaie A, Buresi M, Rao S. Response to Paterson et al. Am J Gastroenterol. 2017;112:1889–92.- Thank you for this reply. We have used a glucose breath test, the information was already provided in our first manuscript. However, we have now highlighted this information and added the Consensus statement as recommended (page 2).
- - principle of operation (company, city, state)
- We thank you for this point. All information requested is now added in Materials and Methods (page 2).
- The North American consensus concluded that “a rise in hydrogen of ≥ 20 ppm
Rezaie A, Buresi M, Lembo A, Lin H, McCallum R, Rao S, et al. Hydrogen and methane-based breath testing in gastrointestinal disorders: the North American consensus. Am J Gastroenterol. 2017;112:775-8- We thank the reviewer for this point. This topic has already been described in detail in our first manuscript and can be found on page 3. The literature was added to the manuscript.
- 7-- what were the exclusion criteria, maybe: proton pump inhibitors, opioids, gastric bypass, colectomy and motility disorders
8-Patients should also fast 8–12 hours before the breath test, avoid smoking the day of the breath test, and minimize physical exertion during the breath test- We apologize for not having addressed these important points raised by the reviewer. We have immediately added the exclusion criteria to our manuscript in detail (page 2, red).
Results
- -are there any patient data on the co-occurrence of diseases: liver, obesity, diabetes, ? what is their relationship to SIBO?
should be presented and discussed- We thank the reviewer for this important comment. We address this point in the discussion and the limitations of our study (page 7). Unfortunately, due to its retrospective character, we are not able to provide detailed information on comorbidities. However, we have now added additional information about IBD patients with positive SIBO on their current medication (when tested positive, table 2B).
discussion
- Chen et al linked IBS with age in their meta-analysis, therefore increasing the impact of SIBO comorbidities is desirable
Chen B, Kim JJ, Zhang Y, Du L, Dai N. Prevalence and predictors of small intestinal bacterial overgrowth in irritable bowel syndrome: a systematic review and meta-analysis. J Gastroenterol. 2018. https://doi.org/10.1007/s00535-018-1476-9.
11. Choung et al found that SIBO appears to be more prevalent in women and in older individuals
Choung RS, Ruff KC, Malhotra A, et al. Clinical predictors of small intestinal bacterial overgrowth by duodenal aspirate culture. Aliment Pharmacol Ther 2011;33:1059–67.- We thank the reviewer for both points and both manuscripts provided. The information was added in the context of our discussion (page 7), both manuscripts were included in our references.
- Limitations of the study should be presented
- We thank for this suggestion. We had this part already included in our manuscript. As requested we have now added further details and limitations of our study in the discussion (page 7).
- there is no conclusion - a separate summary section should be added
- We thank for this important point. We apologize for the missing conclusion in our first manuscript. A conclusion was prepared but unfortunately not transferred obviously. We have of course added this part now to the manuscript (page 7).

Round 2
Reviewer 2 Report
Dear Authors
I am grateful to the authors for their comprehensive answers
best regards
Author Response
We thank this reviewer for this kind response and thank this reviewer for supporting our manuscript to be published in this journal and that no further revisions are necessary.